# Lifing the Effects of Crystallographic Orientation on the Thermo-Mechanical Fatigue Behaviour of a Single-Crystal Superalloy

**DOI:** 10.3390/ma12060998

**Published:** 2019-03-26

**Authors:** Richard Smith, Robert Lancaster, Jonathan Jones, Julian Mason-Flucke

**Affiliations:** 1Institute of Structural Materials, Swansea University, Bay Campus, Swansea SA1 8EN, UK; 640698@swansea.ac.uk (R.S.); jonathan.p.jones@swansea.ac.uk (J.J.); 2Rolls-Royce plc, Bristol BS11JQ, UK; julian.mason-flucke@rolls-royce.com

**Keywords:** thermo-mechanical fatigue, single crystal, CMSX-4^®^, lifing

## Abstract

Thermo-mechanical fatigue (TMF) is a complex damage mechanism that is considered to be one of the most dominant life limiting factors in hot-section components. Turbine blades and nozzle guide vanes are particularly susceptible to this form of material degradation, which result from the simultaneous cycling of mechanical and thermal loads. The realisation of TMF conditions in a laboratory environment is a significant challenge for design engineers and materials scientists. Effort has been made to replicate the in-service environments of single crystal (SX) materials where a lifing methodology that encompasses all of the arduous conditions and interactions present through a typical TMF cycle has been proposed. Traditional procedures for the estimation of TMF life typically adopt empirical correlative approaches with isothermal low cycle fatigue data. However, these methods are largely restricted to polycrystalline alloys, and a more innovative approach is now required for single-crystal superalloys, to accommodate the alternative crystallographic orientations in which these alloys can be solidified.

## 1. Introduction

The maximum operating temperature of turbine and nozzle guide vane materials is of upmost importance for engine manufacturers, since turbine entry temperatures (TET) are critical for the level of performance and efficiency that is achieved during flight. However, during the typical flight cycle of a gas turbine engine, which consists of take-off, cruise, descent and landing, hot section components are also exposed to highly damaging alternating thermal and mechanical loads that act simultaneously. Under such conditions, cracks can potentially nucleate and propagate to failure; a form of damage that is now widely recognised as being thermo-mechanical fatigue (TMF). TMF has been acknowledged to be one of the most severe damage mechanisms facing the aero power generation sector, as designers push materials above and beyond their operating limits, striving to meet the demanding emissions and fuel burn of ACARE ’Flightpath 2050’ targets [1]. 

The publication of ASTM E2368 in 2010 [2], and more recently, ISO12111 in 2011 [3] on strain-controlled TMF standards emphasises the significance of dynamic temperature effects on material fatigue behaviour. It is recognised that the relative contributions of these factors during TMF are a combination of the fluctuating mechanical and thermal conditions. It is now well-documented that TMF is a complex mode of damage, and that there are three separate mechanisms acting simultaneously at high temperatures [4,5]. This includes fatigue, creep and environmental elements; depending on the phase angle (*φ*), the relative contributions of these may vary. The proportions and combinations of each mechanism are controlled by several factors such as peak temperature (*T_MAX_*), temperature range (Δ*T*), applied stress (*σ_MAX_*) and stress range (Δ*σ*). A consequence of the additional variable *φ* is the infinite number of potential TMF conditions; however, a limited number are outlined to be of use [6]. The extent of these mechanisms will also be dependent on the alloy in question, with alloys designed for higher temperature operation such as CMSX-4^®^ being less susceptible [4]. 

One of the more traditional procedures for the estimation of TMF life is empirical correlation with isothermal low cycle fatigue (LCF) data [7,8]. For polycrystalline alloys, this method has been reasonably successful [4,9,10] but in the case of single crystal superalloys, there are many types of intricate TMF failure mechanisms that can contribute to material damage and make direct correlations with isothermal data difficult. In addition, given the strong anisotropic nature of directionally solidified materials, and the preferred <001> growth orientation of many Ni-based superalloys, an added complexity is present when attempting to derive a model for data that is generated from multiple crystallographic orientations. 

Currently, from an engineering standpoint, the standard method for the analysis of cooled turbine aerofoils first involves the prediction of the *T_MAX_* and elastic strain (*ε_El_*)/stress range (Δ*σ*) through an engine cycle. Where there is a significant mechanical load present, it will manifest as additional creep damage, and the predicted fatigue life is downgraded through the use of a non-linear creep-fatigue interaction model. Initiation life can then be calculated by comparing the strain range (Δ*ε*) and peak temperature (*T_MAX_*) to strain-controlled isothermal fatigue (IF) data with a similar loading frequency. Under isothermal testing conditions, *T_MAX_* is the worst-case scenario, but it is commonly reported that the transient temperatures experienced during TMF have a detrimental effect on the life of single crystals [11]. New research has also pointed out that this traditional engineering method may not capture all of the damage mechanisms that take place during TMF of single-crystal materials [12]. There is also circumstantial evidence from comparing life predictions to service and test evidence, that traditional turbine lifing approaches are conservative under certain loading conditions. Therefore, this paper will look at developing an improved understanding of SX behaviour under TMF conditions, and it will provide a more robust predictive model that can provide accurate extrapolation from a known set of results.

## 2. Materials and Methods

CMSX-4^®^ is a second-generation Ni-base single-crystal superalloy. It is currently utilised for turbine blades, due to its elevated temperature capabilities that are routinely used in proximity to the <001> orientation. CMSX-4^®^ was developed by the Cannon-Muskegon Corporation (Muskegon, MI, USA) in the 1990s, and it contains 3 wt % Re and a 70% volume fraction of the coherent γ’ precipitate strengthening phase. The chemical composition of CMSX-4^®^ is detailed in Table 1, and the typical microstructure is illustrated in Figure 1. In this study, research was undertaken on both hot isostatically pressed (HIP) and non-HIP (UNHIPP’D) CMSX-4^®^. The UNHIPP’D material was subjected to a solution heat treatment to dissolve the γ/γ’ eutectic, followed by a gas fan quench. The material was then double-aged at 1140 °C for 2 h and 870 °C for 16 h, to produce a regular cuboidal γ’ microstructure, as depicted in Figure 1. The HIP material was subjected to a similar heat treatment regime, with a prior high-temperature HIP cycle, to significantly reduce casting porosity features. 

## 3. Development of the TMF Lifing Model

Given the complex nature of high-temperature TMF damage, a viable model must consider several different factors, including fatigue, creep and oxidation. As such, the first stage of the model is to modify isothermal fatigue data based on the equivalent isothermal oxidation damage, to produce a universal curve. This is achieved by utilising the exponential rate equation, Equation (1), which incorporates normalised data up to 1050 °C, by altering the time basis to account for relative oxidation performance. The oxidation performance is defined as the time that is taken to reach 0.25 mm oxide penetration.
(1)$$\delta^{.}=Be^{CT}$$
where $\delta^{.}$ = rate of metal loss (mm/hr); *B, C* = constants; *T* = temperature at condition (°C).

The stress state of the isothermal tests is then normalised to $R_{\sigma }$ = −1 using the Walker mean stress (*σ_MEAN_*) correction method, where *m* = 0.6 was found to be the optimal value. The Walker *R* ratio correction method is given in Equation (2), for the stress life correlations: (2)$$∆\sigma_{eq}=\left(\sigma_{max}-\;\sigma_{min}\right)\;\times \;{\left(1-R_{\sigma }\right)}^{\left(m-1\right)}$$
where $∆\sigma_{eq}$ is the equivalent Walker-corrected stress range; $\sigma_{MAX}$ is the maximum stabilised (*N_f_*/2) tensile stress in the cycle; $\sigma_{MIN}$ is the minimum stabilised (*N_f_*/2) tensile stress in the cycle and $R_{\sigma }$ is the $R_{\sigma }$ ratio, a measure of the mean stress in the cycle. 

Figure 2 shows the isothermal data modified to 1050 °C using the oxidation equivalence and Walker *σ_MEAN_* correction method (*m* = 0.6). This approach has produced a suitable degree of correlation for isothermal data at temperatures of 850 °C and above, since at these temperatures, fatigue crack initiation is expected to be driven by oxidation at the specimen surface [13,14,15]. At lower temperatures, fatigue crack initiation transitions to nucleation at subsurface porosity, and it is therefore independent of oxidation [16,17].

From the plot in Figure 2, a universal polynomial curve has been produced, and it provides the basis for the TMF lifing model. Figure 3 shows the isothermal fatigue data normalised to 1050 °C, to account for the relative oxidation performance. The graph illustrates that this is an appropriate means of correlating a wide range of data onto a single curve. The stress state of the material has been corrected to an equivalent $R_{\sigma }$ = −1 condition, using the Walker mean stress correction method, whilst the oxidation factor for 850 °C has been modified by a factor of 2 to produce a more favourable correlation, given the higher levels of expected oxidation. This method provides an improved correlation over the Walker correction to an $R_{\sigma }$ = 0. This is significant, as it makes the R ratio model simpler, since a mean stress correction is only used at tensile conditions above $R_{\sigma }$ = −1. An $R_{\sigma }$ = −1 condition is closer to a “pure” fatigue cycle, as the fully reversed $\sigma_{MAX}$ and $\sigma_{MIN}$ are of a lower magnitude, and therefore less creep damage is present during the highest tensile proportion of the cycle.

Through a review of the historical TMF data, the stabilised stress range (Δ*σ_STAB_*) is recognised as an important correlating parameter and should also be considered in any lifing approach, to produce a more accurate model. Δ*σ_STAB_* occurs at the extreme temperature ends (*T_MAX_* and *T_MIN_*) of standard In-Phase (IP) and Out-of-Phase (OP) cycles. However, due to the complex nature of TMF, and the myriad of possible phase angles, extracting the correct Δ*σ_STAB_* for other cycle types, particularly with a phase shift (<20°), is not as straightforward. In tests where the $R_{\sigma }$ value “shakes down” to a value that is more tensile than $R_{\sigma }$ = −1, the Walker correction method is used with *m* = 0.6, as follows:(3)$$\sigma_{EQ}=∆\sigma_{R-1}=∆\sigma {\left\{\left(1-R\right)/2\right\}}^{\left(m-1\right)}$$

Conversely if the $R_{\sigma }$ value is more compressive than $R_{\sigma }$ = −1 then *Δσ_STAB_* is used:(4)$$\sigma_{EQ}=∆\sigma_{R-1}=∆\sigma ,\;\mathrm{for}\;R_{\sigma }<-1$$

Figure 4 shows the IP TMF results for data generated between Δ*T* = 450–950 °C and Δ*T* = 550–1050 °C, both with and without a 2-min dwell period, and at the same oxidation correction levels for all tests. Here, no additional correction is made for an extended hold period at elevated temperature and strain. Using this method, all of the IP tests at *T_MAX_* of 950 °C and 1050 °C sit within the isothermal line, and they show a more accurate life prediction, which suggests that the length of time that the material is exposed to at elevated temperature is not as important as the stress–temperature combination. Modifying the oxidation correction method to include the dwell period does not produce a favourable fit, so that it is not factored into further analysis. Therefore, a temperature-only correction may be the most successful means of correlating isothermal and TMF lives for tests in which the creep life usage is low. 

This behaviour is similar to that seen in the OP regime. Figure 5 presents OP data for Δ*T* = 550–1050 °C, both with and without a 2-min dwell period, and *ΔT* = 650–1150 °C with no dwell period. The correlation model is applied to the OP data at *T_MAX_* of 1050 °C and 1150 °C, and as can be seen, once again, a dwell period does not seem to add to further damage to the material and as such, a reduction in life is not seen. At 1050 °C, the TMF data sits predominately below the isothermal datum line. The $R_{\sigma }$ ratios for the OP tests at this temperature are mainly between $R_{\sigma }$= 0 and −1, so that $\sigma_{EQ}$ is taken as the Walker-corrected Δ*σ*. 

At 1150 °C, the OP TMF has not been captured well by the model, as the data can be seen to sit well above the universal curve. At longer lives, approaching the endurance limit of the material, the universal curve appears to reach a plateau. Therefore, a more accurate correlation is necessary for these conditions when *Δσ_STAB_* is used. However, the same behavioural trend is not evident at 1050 °C, because generally, the correlation is worse. This may be a consequence of the OP TMF cycle. The peak tensile stress occurs at a low temperature; as a result no creep damage is expected to be present. As discussed previously, the Walker correction method has been shown to account for the difference in creep damage between fatigue cycles at different *σ_MEAN_* values. Therefore, OP TMF conditions at these elevated temperatures may exceed the applicability of the Walker correction method. 

## 4. Effects of Orientation on TMF Life

### 4.1. In-Phase TMF

With the foundations of the model now established, the next factor to consider is the effect of primary crystallographic orientation and whether this can be correlated. Figure 6 presents IP CMSX-4^®^ data for the three alternative crystallographic orientations, <001>, <011> and <111> for Δ*T* = 450 °C–950 °C and Δ*T* = 550 °C–1050 °C. As shown, the model successfully captures the effects of the <011> and <001> orientations at peak cycle temperatures of 950 and 1050 °C, as all the data points apart from the unbroken (ub) sample sit within a factor of 2*N_f_* of the isothermal datum curve. In contrast, the <111> results are not suitably captured by the TMF model. In addition, a dwell period seems to reduce the life, but it increase the Δ*σ* of the <111> orientated IP TMF tests. As such, additional parameters need to be considered for this particular orientation. 

### 4.2. Out-of-Phase TMF

In a similar manner to the IP model, Figure 7 displays the effect of primary orientation on OP TMF. Here, it appears that the only data that consistently fits within the universal curves is the <001> data under a 2-min dwell period at 1050 °C. However, the model breaks down when trying to capture all alternative crystallographic orientations, both with and without a hold period. One of the possible reasons for this behaviour could be the complex nature of the strain-temperature relationship, otherwise known as phase angle. Previously, reporting TMF at various intermediate phase angles by using Δ*σ_STAB_* and *T_MAX_* was observed to produce conservative lifetime predictions. This may be a result of a more damaging combination of *Δσ* and temperature during the cycle. Therefore, extracting the correct *Δσ* for these tests is of upmost importance.

Figure 8 shows a schematic representation of Δ*σ_STAB_* and the improved Δ*σ* calculation used for the phase shift tests. As Δ*σ_STAB_* (A) provides a conservative prediction, a new means of reporting *Δσ* was required for intermediate phase angles, such as the 20° phase shift (PS) TMF results. The *Δσ* can be taken at either the intermediate temperature point, i.e., 875 °C (B) or *T_MAX_*, i.e., 950 °C (C). The figure illustrates how the different means of extracting the *Δσ* for phase shift TMF results correlate with the isothermal datum curve. The model does not capture the behaviour of phase shift TMF orientated in the <001> and <111> directions when based on the equivalent stress ($\sigma_{EQ}$), which is also the case for *Δσ_STAB_* as all the $R_{\sigma }$ values for all tests are more tensile than $R_{\sigma }$ = −1. Therefore, using the methods described, an improved approach is defined. The $\sigma_{EQ}$ using the stress at *T_MAX_* of 950 °C (C) which does not account for $\sigma_{MAX}$ at point A, works well for <001>, but less so for <111>. In this calculation, $\sigma_{MIN}$ is still used to calculate the range with the stress at 950 °C. The discrepancy in the <111> data could be attributed to a stronger response in this orientation whilst under TMF loading conditions, an effect that is not observed in the isothermal data. An important factor to consider here is the higher Young’s modulus that is typically seen in the <111> orientation in comparison to the other directions of growth, and the potential benefit in TMF performance that this may offer. The equivalent stress ($\sigma_{EQ}$) using the stress at an intermediate temperature of 875 °C (B), which once again does not account for the $\sigma_{MAX}$ at point A, gives a lower life and works well for <111>, but less so for <001>. This may be caused by an issue with the “temperature” factor that occurs at low temperatures. The intermediate temperature point (875 °C) sits close to the transition temperature of fatigue crack initiation, which has been discussed in previous work [15,18]. With the consideration of this factor, Figure 9 has been derived to more accurately replicate the stress behaviour in specimens that are subjected to a phase shift. The figure shows that the stress at *T_MAX_* provides the best fit of data, particularly when considering that the <111> data are stronger than the <001> results. A similar characteristic trend is observed for <111>.

Figure 10 shows the −135° OP TMF data for *ΔT* = 350 °C–1050 °C, plotted using the original TMF lifing model derived for IP and OP TMF. As expected, the Walker correction of the full *Δσ* does not provide a suitable correlation. This discrepancy is expected, since the −135° OP strain-temperature behaviour experiences a higher stress at a lower temperature, which is not as damaging as the simultaneous peak stress and peak temperature combinations seen in IP conditions. In addition, an $R_{\sigma }$ correction model to compensate for *σ_MEAN_* effects (creep) would not be required for a TMF cycle where $\sigma_{MAX}$ is under low-temperature conditions. Plotting Δ*σ* alone provides a closer correlation, but this is still not acceptable, although the scatter exhibited by −135° OP TMF appears to be reduced.

Figure 11 shows −135° OP TMF plotted using Δ*σ* at peak cycle temperature (C). This combination provides an excellent correlation with the isothermal data, and the tensile and compressive data form a single population. This is consistent with the results seen for PS TMF <001>. The stress data points have been estimated from the maximum and minimum values from the cycle, as no loop data is available. Also, no intermediate calculations have been carried out to determine whether a lower stress–temperature combination can be calculated.

Figure 12 shows the final model, fitted to all of the <001> and <011> orientated data. The data has been modified by altering the time basis to account for the relative oxidation performance. The results have been normalised to 1050 °C, and the oxidation factor at 850 °C has been reduced twice, to account for the transition in fatigue crack initiation observed at lower temperatures. As a dwell period has been observed to have no effect, the oxidation correction has been modified to account for temperature only, and not the time spent at the temperature. Also, the baseline for isothermal tests has been altered from 4 s to 2 s to account for this, as the effect of oxidation is predominantly based on the exposure temperature, rather than on the time spent at that temperature. All intermediate phase angle tests including the 20° phase shift and −135° OP TMF tests, have been plotted using the *Δσ* at *T_MAX_*. The model works well across all test types, with additional LCF results being included for comparative purposes. The $R_{\sigma }$ ratio model is used to determine the equivalent stress values that are used for each test. At $R_{\sigma }$ below < −1, *Δσ_STAB_* is used, as the compressive condition does not incur significant creep damage, so that a *σ_MEAN_* correction method is not required. Conversely, for a more tensile $R_{\sigma }$ above > −1, a Walker correction is required, using an *m* = 0.6. 

Figure 13 shows the model applied to the <111> orientated CMSX-4^®^ data. An increase in strength is observed, and the model does not seem to capture this single crystal orientation. However, the <111> orientated material does follow a similar curve to the <001> data, but at a higher strength. 

## 5. Conclusions

The TMF performance of CMSX-4^®^ has been investigated through a review and meta-analysis of historical TMF and isothermal fatigue data. Analysis has shown that it is possible to successfully and accurately correlate multiple TMF test conditions with isothermal low cycle fatigue tests carried out at 1050 °C, using the TMF lifing methodology outlined in this paper. The TMF lifing methodology is based upon the exponential rate equation, which corrects the TMF life based on the rate of oxide penetration at peak cycle temperature. For the in-phase and out-of-phase TMF carried out at peak cycle temperatures between 950 °C–1050 °C, the Walker mean stress correction method provides a meaningful correlation. However, for the more complex phase angles that include a 20° phase shift and −135° OP, an improved correlating parameter is required. Extracting the stress range at peak cycle temperature provides beneficial results for these intermediate-phase angles, but a more substantial data set comprising high quality mechanical results for all prominent phase angles could enhance the robustness of this universal model even further.

## Figures and Tables

**Figure 1 materials-12-00998-f001:**
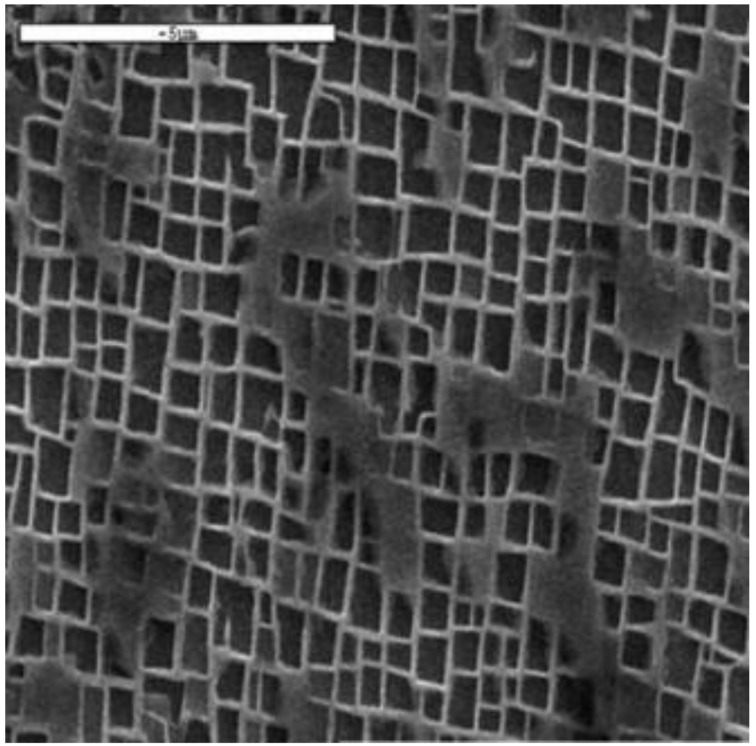
Scanning electron microscopy (SEM) micrograph of the CMSX-4^®^ microstructure, showing cuboidal γ’ precipitates surrounded by the primary γ matrix. Reproduced from [11].

**Figure 2 materials-12-00998-f002:**
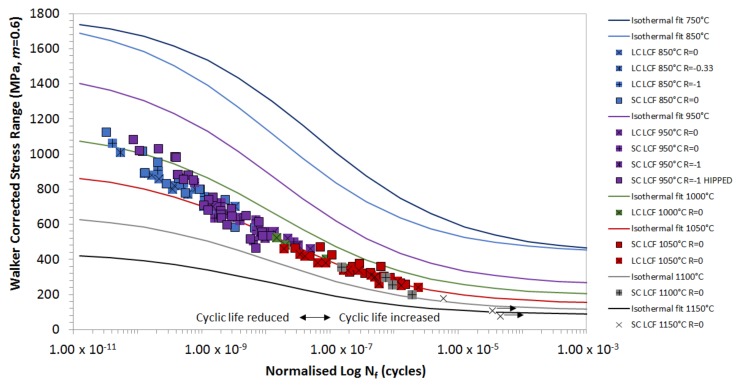
Correlation of low cycle fatigue (LCF) data, using oxidation correction factors to modify life, depending on relative oxidation performance.

**Figure 3 materials-12-00998-f003:**
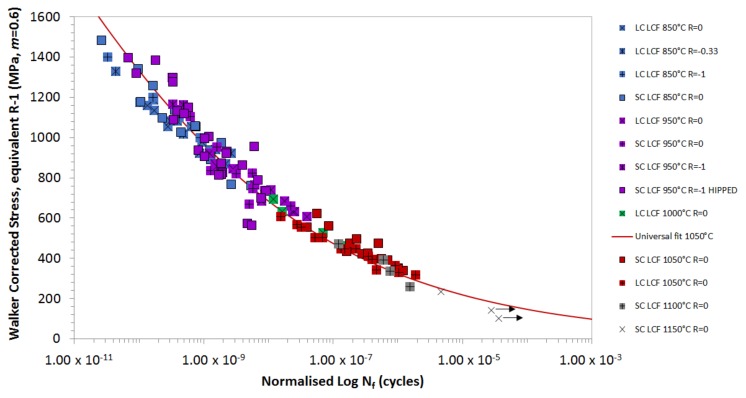
Isothermal fatigue data normalised to 1050 °C to account for relative oxidation performance, and used to produce a universal LCF curve.

**Figure 4 materials-12-00998-f004:**
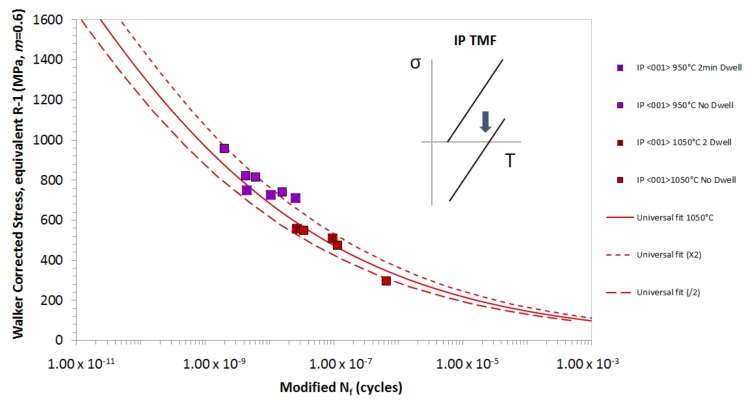
A thermo-mechanical fatigue (TMF) lifing model is applied to the IP TMF results for Δ*T* = 450–950 °C and Δ*T* = 550–1050 °C, not including the 2 min dwell period data.

**Figure 5 materials-12-00998-f005:**
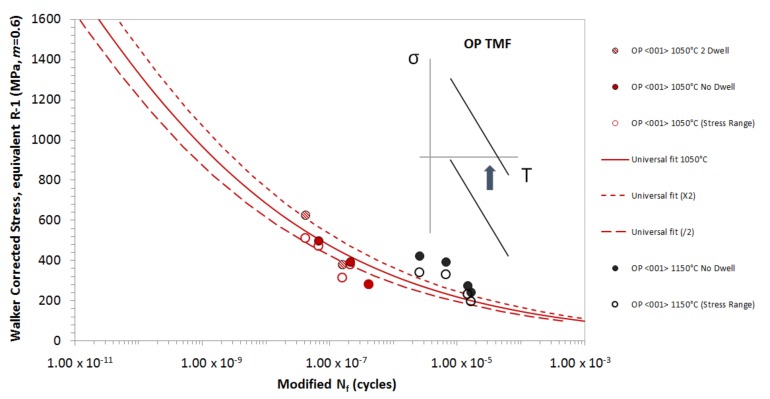
TMF lifing model applied to OP TMF results for Δ*T* = 550–1050 °C, and Δ*T* = 650–1150 °C.

**Figure 6 materials-12-00998-f006:**
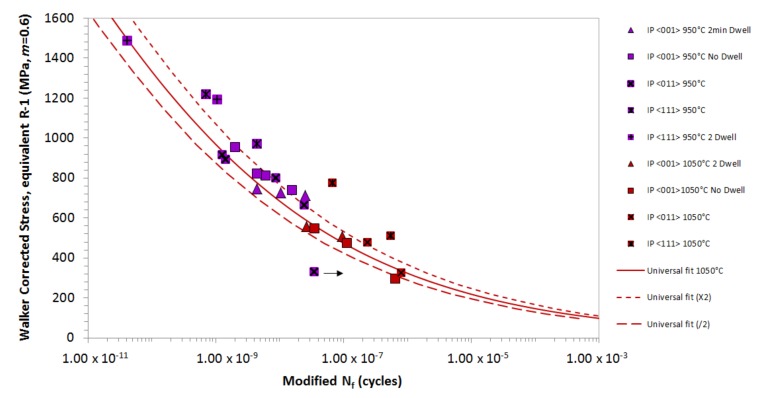
Effect of a single crystal growth orientation on in-phase TMF life.

**Figure 7 materials-12-00998-f007:**
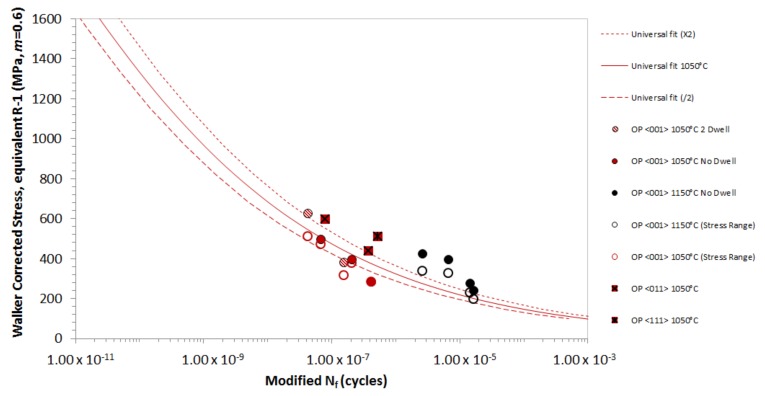
Effect of single crystal growth orientation on out-of-phase TMF life.

**Figure 8 materials-12-00998-f008:**
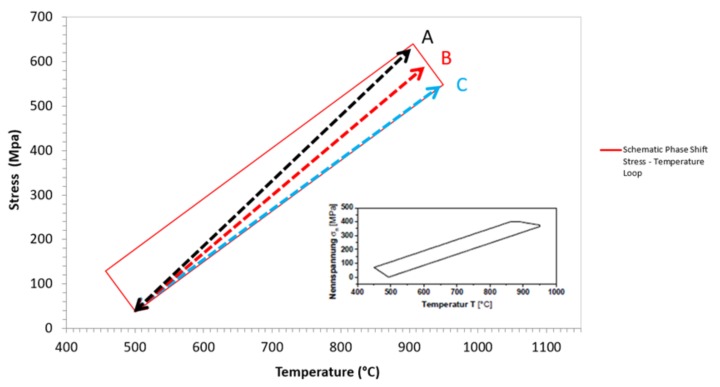
A schematic representation of *Δσ_STAB_* and the improved *Δσ* calculation used for the <20° phase shift tests.

**Figure 9 materials-12-00998-f009:**
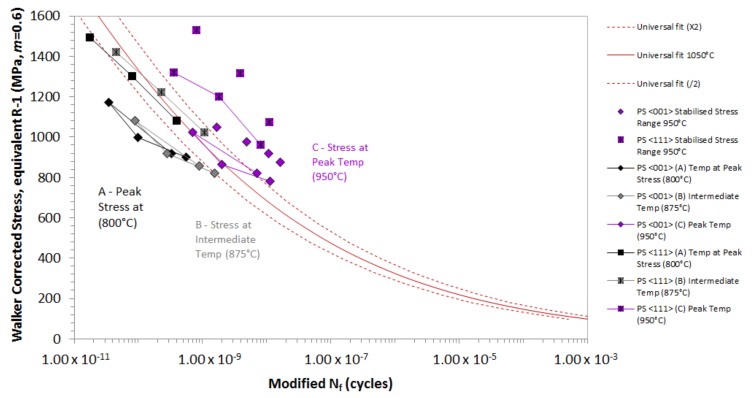
Illustration of how the different means of extracting the stress range for the 20° phase shift TMF correlate with the isothermal datum curve.

**Figure 10 materials-12-00998-f010:**
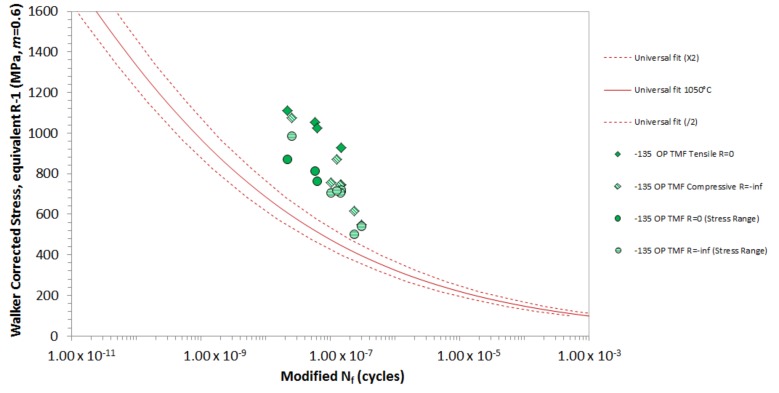
−135° OP TMF for *ΔT* = 350 °C–1050 °C, plotted using the original TMF lifing model used for IP and OP TMF.

**Figure 11 materials-12-00998-f011:**
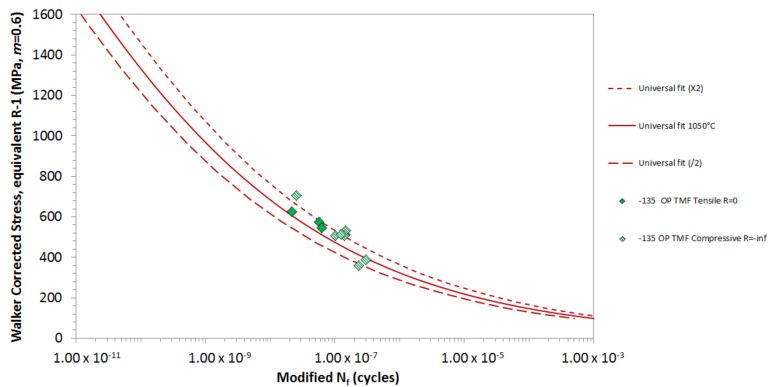
−135 OP TMF, plotted using the stress at *T_MAX_*.

**Figure 12 materials-12-00998-f012:**
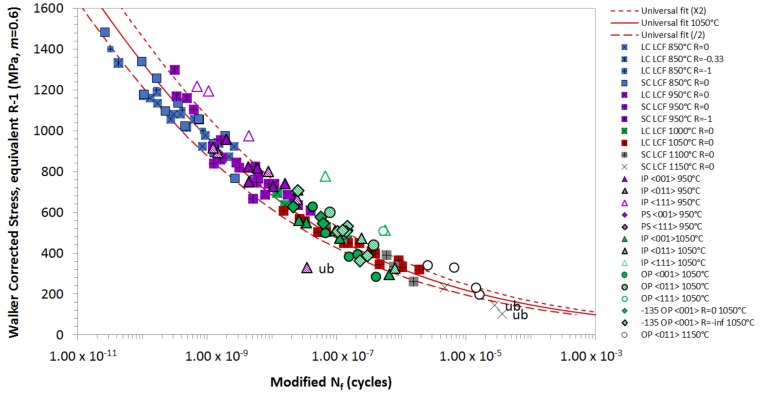
Final TMF model fitted to all the <001> and <011> orientated CMSX-4^®^ data.

**Figure 13 materials-12-00998-f013:**
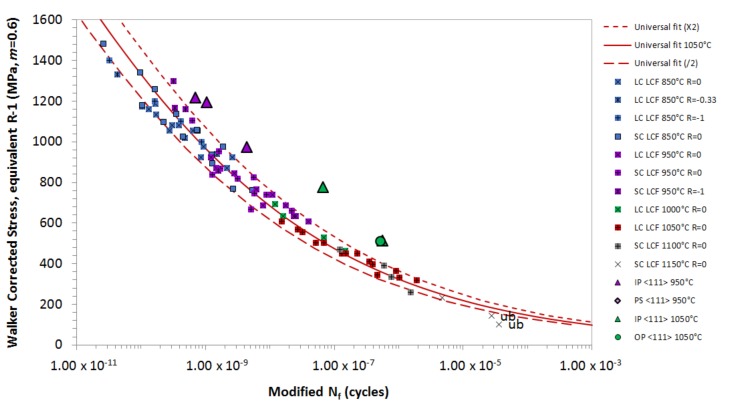
Final TMF model fitted to all the <111> orientated CMSX-4^®^ data.

**Table 1 materials-12-00998-t001:** The Chemical composition of CMSX-4^®^ (wt %).

Alloy	Cr	Co	Mo	W	Ta	Re	Nb	Al	Ti	Hf	Nl	Density
CMSX-4	6.4	9.6	0.6	6	6.5	3	-	5.6	1	0.1	Bal.	8.70

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
