# Peer review of "Lifing the Effects of Crystallographic Orientation on the Thermo-Mechanical Fatigue Behaviour of a Single-Crystal Superalloy"

_materials, 2019, doi:10.3390/ma12060998_

Round 1
Reviewer 1 Report
it will be very interesting to give a short and explicite summary to describe
- how the in-phase and the out-of-phase TMF data have been corrected numerically with the used model,
- the associated improvement and the divergence of corrected data;
- the possible errrors and uncertainty with the used of model.
Author Response
1. It will be very interesting to give a short and explicite summary to describe how the in-phase and the out-of-phase TMF data have been corrected numerically with the used model
The authors feel that the paper is quite explicit on what was being done to the data e.g. eqn 3 & 4 + fig 4 & 5 show what is happening to the stress state in a general manner.
2. it will be very interesting to give a short and explicite summary to describe the associated improvement and the divergence of corrected data;
In the authors’ opinion, the best way would be to show corrected and uncorrected data visually but this would uncover the original data which would be difficult to clear for publication due to it’s industrial sensitivity.
3. it will be very interesting to give a short and explicite summary to describe the possible errrors and uncertainty with the used of model.
The model presented in this paper is to provide a proof of concept framework of a model that could potentially be used by industry (around TRL3). Some aspects will work "out-of-the-box" e.g. IP TMF behaviour. OP TMF behaviour & SX effects require more work but provided that the user is aware of the uncertainty you can utilise in design (but would be designing with conservatism). Clearly a substantial validation effort would be required before actual use which would be required for any new method but this may require an industry / academic -wide effort to proof this.
Reviewer 2 Report
The submitted, high niveau manuscript entitled ‘Lifing the Effect of Crystallographic Orientation on the Thermo-Mechanical Fatigue Behaviour of a Single Crystal Superalloy’ deals with the evaluation of historical TMF test results and isotehermal fatigue data of a singlecrystal material. The manuscript is winteresting and worth to publish, during its careful manuscript only a few concerns arose.
- How was the chemical composition in Table 1 measured?
- Please identify the phases in fig. 1 by labels.
- The most important concern of this Reviewer is in the reliability of the modified and corrected data. Could the Authors address this issue? Are these data suitable and reliable for turbine blades design for example?
Author Response
1. How was the chemical composition in Table 1 measured?
This is the standard composition taken from the Cannon-Muskegon website, the original manufacturers of the material. As such, the authors do not see the need to add this information to the paper
2. Please identify the phases in fig. 1 by labels.
This information is already given in the figure caption for Figure 1. The word ‘cuboidal’ has been added to aid clarity
3. The most important concern of this Reviewer is in the reliability of the modified and corrected data. Could the Authors address this issue? Are these data suitable and reliable for turbine blades design for example?
The model presented in this paper is to provide a proof of concept framework of a model that could potentially be used by industry (around TRL3). Some aspects will work "out-of-the-box" e.g. IP TMF behaviour. OP TMF behaviour & SX effects require more work but provided that the user is aware of the uncertainty you can utilise in design (but would be designing with conservatism). Clearly a substantial validation effort would be required before actual use which would be required for any new method but this may require an industry / academic -wide effort to proof this.
Reviewer 3 Report
It is an interesting and curious paper with well-organised text and legible content. It brings a new approach to the analysis of thermo-mechanical fatigue and lifing of single-crystal superalloys.
Author Response
No comments to address